# OperonSEQer: A set of machine-learning algorithms with threshold voting for detection of operon pairs using short-read RNA-sequencing data

Raga Krishnakumar[1]*, Anne M. Ruffing[2]

1 Systems Biology Department, Sandia National Laboratories, Livermore, California, United States of America, 2 Molecular and Microbiology Department, Sandia National Laboratories, Albuquerque, New Mexico, United States of America

* rkrishn@sandia.gov

**Data Availability Statement:** Code availability: OperonSEQer is available at https://github.com/sandialabs/OperonSEQer.

**Funding:** This work was supported by the Laboratory Directed Research and Development

## Abstract

Operon prediction in prokaryotes is critical not only for understanding the regulation of endogenous gene expression, but also for exogenous targeting of genes using newly developed tools such as CRISPR-based gene modulation. A number of methods have used transcriptomics data to predict operons, based on the premise that contiguous genes in an operon will be expressed at similar levels. While promising results have been observed using these methods, most of them do not address uncertainty caused by technical variability between experiments, which is especially relevant when the amount of data available is small. In addition, many existing methods do not provide the flexibility to determine the stringency with which genes should be evaluated for being in an operon pair. We present OperonSEQer, a set of machine learning algorithms that uses the statistic and p-value from a non-parametric analysis of variance test (Kruskal-Wallis) to determine the likelihood that two adjacent genes are expressed from the same RNA molecule. We implement a voting system to allow users to choose the stringency of operon calls depending on whether your priority is high recall or high specificity. In addition, we provide the code so that users can retrain the algorithm and re-establish hyperparameters based on any data they choose, allowing for this method to be expanded as additional data is generated. We show that our approach detects operon pairs that are missed by current methods by comparing our predictions to publicly available long-read sequencing data. OperonSEQer therefore improves on existing methods in terms of accuracy, flexibility, and adaptability.

## Author summary

Bacteria and archaea, single-cell organisms collectively known as prokaryotes, live in all imaginable environments and comprise the majority of living organisms on this planet. Prokaryotes play a critical role in the homeostasis of multicellular organisms (such as animals and plants) and ecosystems. In addition, bacteria can be pathogenic and cause a

(LDRD) program at Sandia National Laboratories, a multi-mission laboratory managed and operated by National Technology and Engineering Solutions of Sandia, LLC, a wholly owned subsidiary of Honeywell International, Inc., for the U.S. Department of Energy's National Nuclear Security Administration under contract DE-NA0003525. The funders had no role in study design, data collection and analysis, decision to publish, or preparation of the manuscript. AMR is the recipient and PI of the LDRD under which the work was conducted (Project #212957).

**Competing interests:** The authors have declared that no competing interests exist.

variety of diseases in these same hosts and ecosystems. In short, understanding the biology and molecular functions of bacteria and archaea and devising mechanisms to engineer and optimize their properties are critical scientific endeavors with significant implications in healthcare, agriculture, manufacturing, and climate science among others. One major molecular difference between unicellular and multicellular organisms is the way they express genes–multicellular organisms make individual RNA molecules for each gene while, prokaryotes express operons (i.e., a group of genes coding functionally related proteins) in contiguous polycistronic RNA molecules. Understanding which genes exist within operons is critical for elucidating basic biology and for engineering organisms. In this work, we use a combination of statistical and machine learning-based methods to use next-generation sequencing data to predict operon structure across a range of prokaryotes. Our method provides an easily implemented, robust, accurate, and flexible way to determine operon structure in an organism-agnostic manner using readily available data.

## Introduction

Bacteria often transcribe functionally related genes not as single units but as contiguous RNA molecules (i.e., operons)—these molecules are under the control of a single promoter, allowing them to be co-expressed when required [1–6]. Prior to the advent of genomics, operon structure in prokaryotes was empirically determined, starting with the famous paper by Jacob and Monod outlining the structure of the lac operon [6]. Over the decades, more and more operons were identified using reverse transcription (RT)–PCR and recombinant DNA techniques [7–10]. In the 2000s when genomic analyses grew exponentially, a number of newly characterized features of bacterial genomes were used to determine operon structure more globally. An important factor that enhanced operon prediction was conservation of protein product function and gene ordering/distances. A number of studies have used conservation to greatly improve our understanding of operon structure across prokaryotes [11–15]. Other critical features that were considered and shown to affect operon membership were intergenic distance (with shorter distances between genes correlating strongly with operon membership) and the prediction of intrinsic terminators [16]. Finally, demonstrating co-expression of genes using genomic techniques such as microarrays and sequencing was also a critical piece of evidence used to strengthen operon prediction techniques [13,16–21].

Existing operon predictions often show high precision and accuracy for well-annotated organisms, but the fact that many of them require information about gene function and conservation for this accuracy is a caveat [11,13,22,23]. Newer methods include the use of visual representations of the genome to categorize operons [24].

Existing studies using RNA-seq to augment operon predictions demonstrated the usability of RNA-seq data in this context, but there is still a gap in the technology with respect to software that is both broadly-applicable across experimental conditions and species, but also allows the user to decide whether catching the highest number of operon pairs (high recall) or being very discerning (high precision) is most important. Existing operon prediction tools also lack the flexibility to incorporate data from disparate sources with similar reliability, regardless of the organism, experimental conditions or depth of data. We believe that an approach that leverages not raw signal in RNA-seq data (which is highly variable and prone to batch effects), but rather uses statistics to determine the distribution of signal across two genes and an intergenic region provides a broader approach to operon prediction that can be used across a range of data sets and species. In addition, using multiple methods and tallying the results gives the

opportunity for a voting system that can give the user flexibility in what they decide to call a relevant operon pair. It is also increasingly clear that careful characterization of the resulting predictions against long-read-confirmed operons is necessary to truly evaluate the performance of a model, which is a technological opportunity that has recently arisen. Since novel data will continue to be generated using both long- and short-read sequencing, it is necessary to provide the code to re-train and re-evaluate any method developed as this novel data emerges. To continue the work established by these studies and show that individual RNA-seq experiments can be sufficient for operon calls, we developed an operon prediction method, trained using a range of RNA-seq data from different organisms with a range of GC-content, to predict operon structure from a single set of RNA-seq data for two adjacent genes from data that has never been seen by the algorithm. Our approach addresses the issue of variability between RNA-seq data sets without requiring two or more matched experimental conditions, or any information about gene function, thereby building on and advancing the current state of the art in operon prediction. Our method also seeks to address the challenge of normalizing and featurizing the sequencing data to make it generalizable across experiments without any prerequisites.

OperonSEQer uses a non-parametric statistical test (to avoid making assumptions about the data distribution) to obtain the likelihood that the RNA-seq signal coverage across two genes and the intergenic region come from the same distribution. Our hypothesis is that the result of this statistical test, along with intergenic distance, is accurately predictive of an operon pair from any short-read RNA-seq data set, and we demonstrate this using a set of machine learning algorithms trained on existing data. We also show that using this method to identify operons in previously unseen organisms and data sets does not significantly reduce the accuracy, while leaving open the possibility to train the models with additional data sets if necessary. We evaluate six different algorithms and show that while specificity and recall vary for each algorithm, they all perform on-par with existing operon prediction methods. By taking advantage of a multi-algorithm method that uses a threshold voting system, we further improve on this performance. In addition, we show that OperonSEQer identifies new operon pairs that are not found in previous standard predictions but are likely to be true operons based on empirical evidence from previously published long-read *E.coli* RNA-seq data [25]. Finally, we demonstrate that while OperonSEQer can call operons based on a single data point (without replicates) of a gene pair and the intergenic region, having 2 or more replicates per gene pair greatly increases its performance. In summary, our operon calling method matches the state of the art in operon prediction by determining operon status of gene pairs with high precision and recall and advances the state of the art by identifying new operon pairs and by providing flexibility to the user to determine whether they want their results to favor higher recall (i.e. catch every single operon pair) or higher specificity (i.e. make sure anything called is a true positive).

## Results

### Statistical analysis of features from RNA-seq data for operon prediction

The main aims of OperonSEQer are to predict operon status from an arbitrary number of data sets to produce a comprehensive list of potential operons, for these predictions to be statistically robust despite only having a single data set, and to be species-agnostic. While we acknowledge that there are species-specific differences that may affect the outcome of such an algorithm (e.g., intergenic distances are of different lengths in different organisms), our premise was that each two-way comparison of adjacent genes on the same DNA strand, regardless of any other features, was an individual data point and that a range of algorithms could be

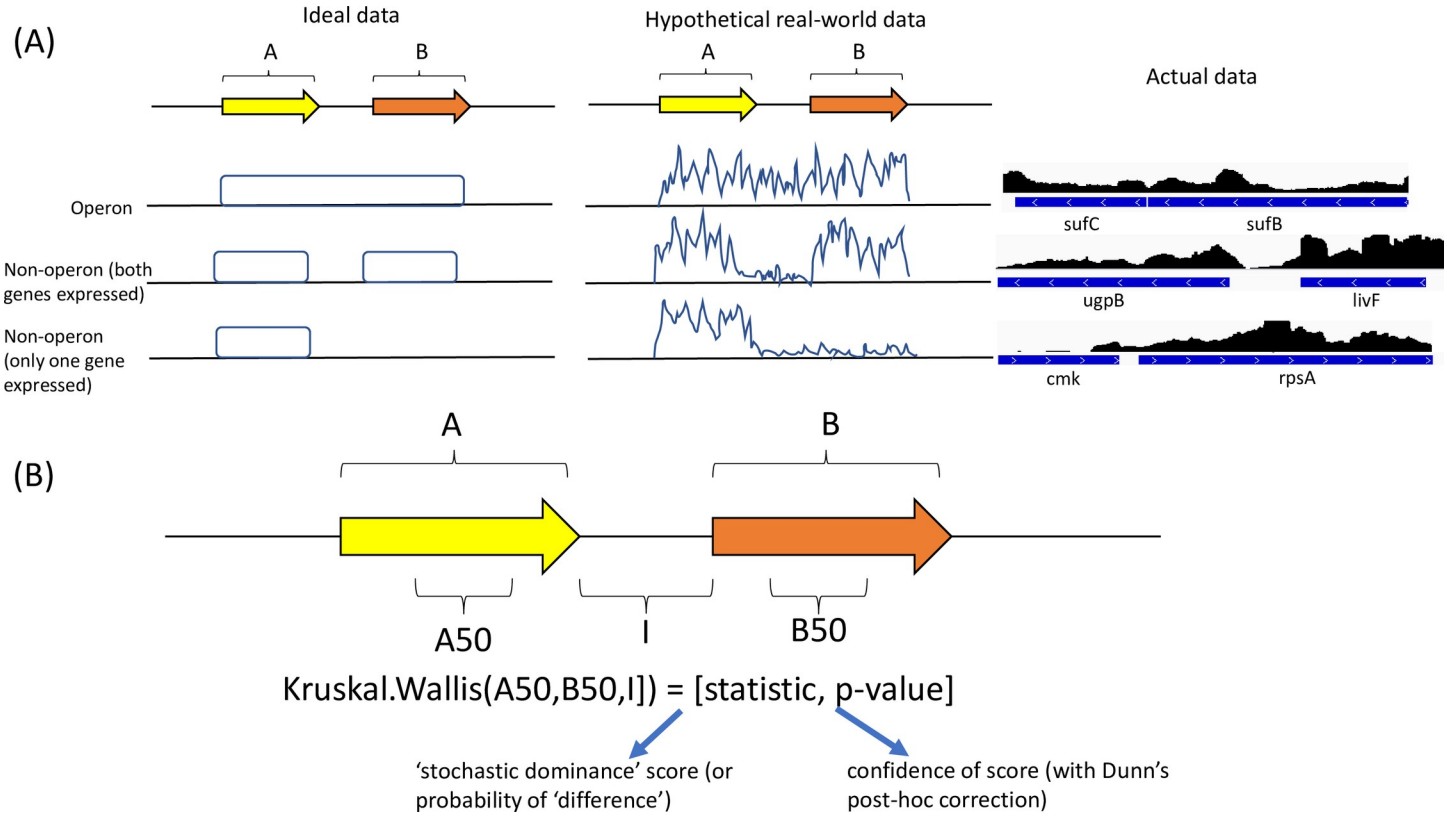

**Fig 1. Schematic of our method for determining similarity of RNA-seq signal between two adjacent genes.** (A) Identification of an operon pair requires at least one of the two genes to be detectably expressed, and significant signal in the intergenic space. Idealized data on the left, hypothetical real-world data in the middle, and actual data on the right. (B) Usage of the Kruskal-Wallis statistic and p-value for pairwise comparisons of genes A, B and the intergenic (I) region, as well as the 3-way comparison. A50 and B50 represent the 50bp from genes A and B that are 50bp away from the intergenic region. These were used for comparison to minimize incorporation of technical variability seen across the gene body. These values, along with the intergenic distance, serve as features for training our operon prediction model.

trained on a compilation of such data points across species, conditions, and replicates. This also allowed us to have many more data points than if we had taken a gene-specific approach. To this end, we established a statistical method that determines whether the RNA-seq coverage signal across the intergenic-flanking regions of two adjacent genes on the same strand is from a single distribution. Using RNA-seq signal from the gene regions directly flanking the intergenic region, as well as the intergenic region itself, a non-parametric rank test (Kruskal-Wallis) was applied to obtain both a statistic and p-value for the comparison of the coverage signal at the three regions–gene A, gene B and the intergenic region (Fig 1). Previous reports have shown that intergenic distance is an important factor in determining whether two genes belong to the same operon, so we used the intergenic distance as well as the Kruskal-Wallis statistic and p-value as features for calling operon gene pairs [16,26].

A challenge in using RNA-seq data to model operons, especially when users do not have the computational resources with bandwidth to train algorithms on enormous amounts of data, is having enough diversity in the input data to cover a wide range of conditions that might be relevant to your organisms of interest. Therefore, OperonSEQer was trained on a wide range of organisms and was designed to allow for user input of additional organism and RNA-seq data for customization and iterative improvement. We used publicly deposited RNA-seq data sets from 7 different bacterial species (both Gram-positive and Gram-negative as well as heterotrophic and photoautotrophic): *Burkholderia pseudomallei* (*B. pseu*), *Clostridium difficile* (*C. diff*),

*Escherichia coli* (*E. coli*), *Synechococcus* sp. PCC 7002 (*Syn. 7002*), *Synechocystis sp*. PCC 6803, *Synechococcus elongatus* PCC 7942 (*S. elon*), *Staphylococcus aureus* (*S. aure*) and *Bacillus subtilis* (*B. subt*) [27–42]. The data were processed and annotated as outlined in the Materials and Methods section, using standard pipelines and publicly available software. In addition, we downloaded standard operon predictions by finding common operon calls between MicrobesOnline and ProOpDB where available [11,13]. Operon predictions from these online tools agreed to a high degree (83% agreement), and therefore, we chose the MicrobesOnline prediction as ground truth for operon structure, as this database had the largest number of organisms. Briefly, MicrobesOnline uses the following criteria to determine their operon pairs: (i) intergenic distance, (ii) conservation, (iii) correlated expression if available, (iv) gene ontology and (v) phylogenic classification [12]. We chose not to combine existing operon calls for *E. coli* since that would skew the accuracy of *E.coli* over other organisms and therefore the skew the trained models.

We performed a correlation analysis to determine the probability that a pair of genes (gene A and gene B with intermediate region I) is in an operon using a number of important features from Kruskal-Wallis (KW) analysis of the RNA-seq data (Fig 2). The features used were: Kruskal-Wallis statistic and Kruskal-Wallis p-value (all 2-way comparisons plus the 3-way comparison) and intergenic distance. The Kruskal-Wallis test was conducted using the 50bp from genes A and B that are adjacent to the intergenic region. This was done to minimize incorporation of technical variability seen across the gene body, which could introduce uninformative noise into the data (Fig 1). A large KW statistic represents a large difference in signal between the groups being compared, and a small p-value indicates that this difference is significant. Using the 2-way and 3-way comparisons, we get 8 dimensions of information, and while it is possible that each of these is uniquely impactful in defining an operon, we acknowledge that some of them may be related (e.g. the 3-way comparison is likely to correlate with 2-way comparisons). Nevertheless, we include all these parameters in our analysis to maximize information use. We used a log10 transformation for the KW p-values to improve resolution. As expected, the length of genes A and B do not correlate with operon structure, and as previously reported [16,26,43,44], intergenic distance correlates negatively with likelihood of an operon pair (Fig 2). In terms of gene expression, the KW statistic correlates negatively with operon pair likelihood, and the log value of the KW p-value correlates positively (Fig 2). Despite RNA-seq data coming from different organisms and disparate sources, we find that the KW statistic and p-value have a higher correlation with operon pairs than intergenic distance, highlighting the importance of the information coming from RNA-seq across species. In addition, metrics that assay RNA-seq coverage of the intergenic region are the most predictive of operon pairs as expected. However, no single data point had higher than 50% correlation, suggesting that inferring a direct linear relationship between any features and the outcome of being in an operon would be too simplistic, therefore requiring a more complex model.

## OperonSEQer improves recall and specificity for operon prediction

To improve operon prediction from RNA-seq data, we used intergenic length, KW statistics, and KW p-values as features for machine learning. We tested a range of classification algorithms that have previously been used in similar applications: logistic regression (LR), support vector machine (SVM, using the radial basis function which we determined to perform better than the linear, sigmoid or polynomial kernels), random forest (RF), XGBoost (XGB) and Gaussian Naïve Bayes (GNB). We used all of the data sets outlined in the methods and initially validated the various models using 50 random bootstraps of 75% of the data for training and 25% of the data for validation [45–48]. Recall and specificity served as measures of success to

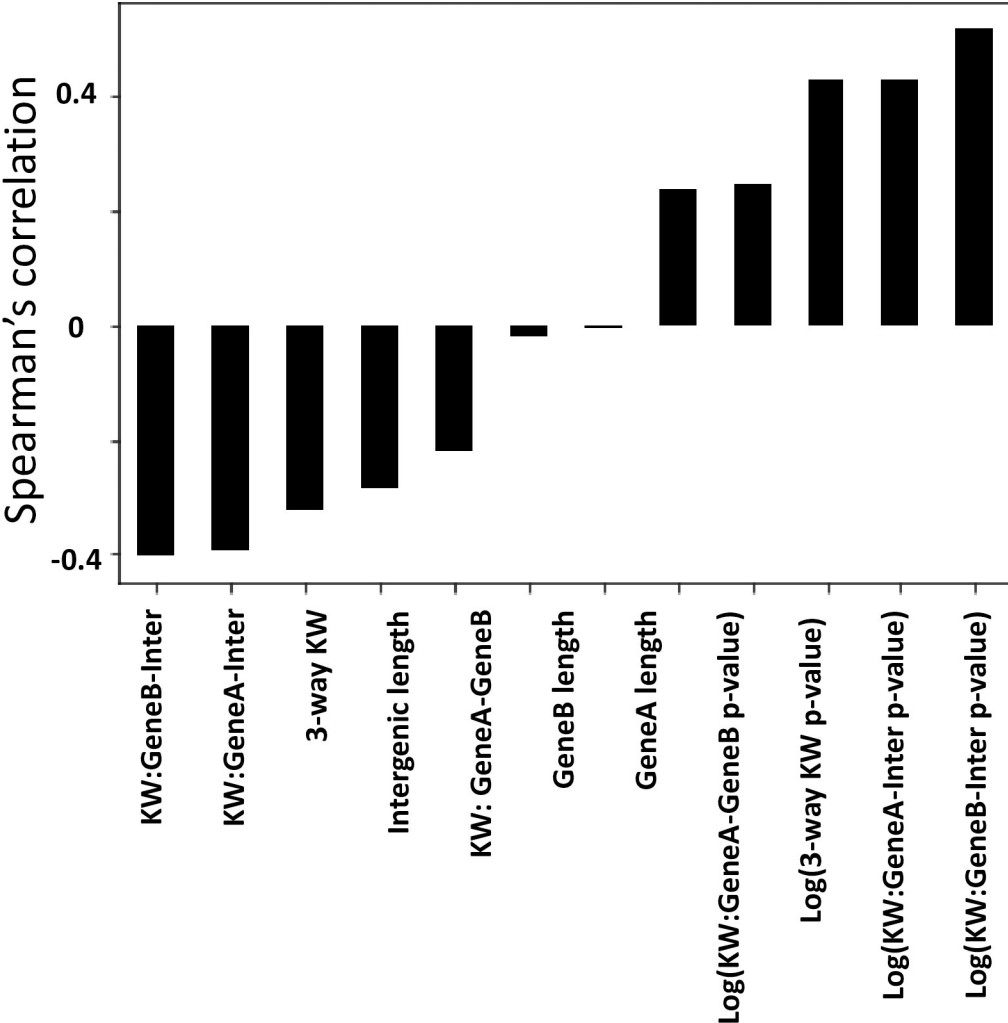

**Fig 2. Operon-SEQer features and performance across the various algorithms used.** (A) Spearman's correlation coefficients between the features considered for use in machine learning and operon pair calls made by MicrobesOnline across 7-species (see main text). KW = Kruskal Wallis statistic. P-value adjusted using Dunn's post-hoc correction.

match previous reports [18,23]. As we are aiming for a species- and gene-agnostic method, these results are an aggregate of all the species and data sets that we included in our analysis.

While there was some trade-off between recall and specificity, all algorithms performed with both recall and specificity of at least 80% (Table 1). In particular, the tree-based methods

**Table 1. Recall and specificity for the validation set for OperonSEQer across six different algorithms.** Heat map colors range from yellow (lowest) to white (mid-point) to blue (highest).

| Algorithm | Recall | Specificity |
|---|---|---|
| Support Vector Machine | 0.91 | 0.84 |
| Multilayer Perceptron | 0.92 | 0.81 |
| Logistic Regression with Ridge | 0.93 | 0.87 |
| Random Forest | 0.95 | 0.94 |
| **Gaussian Naïve Bayes** | 0.95 | 0.80 |
| XGBoost | 0.99 | 0.99 |

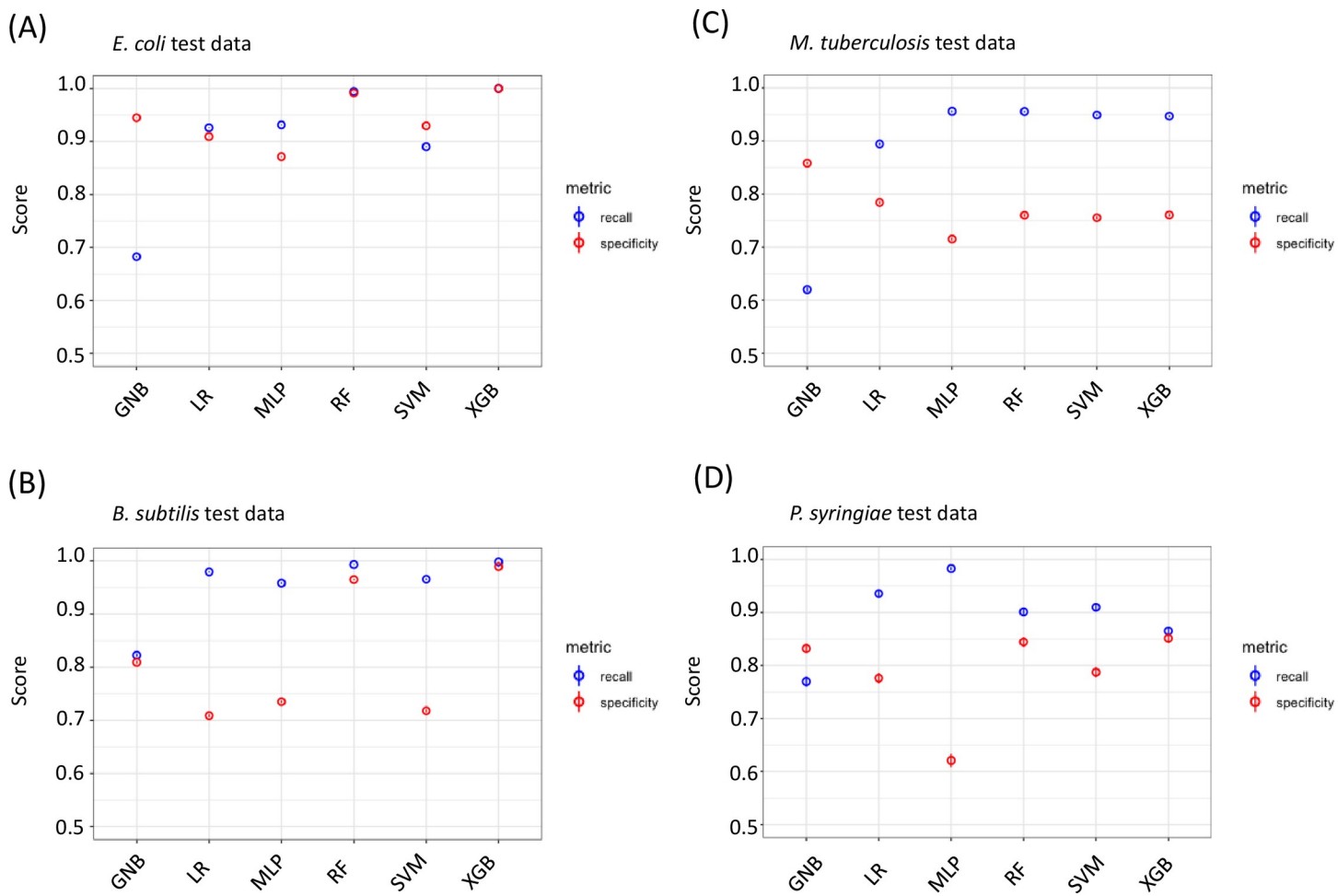

**Fig 3. OperonSEQer can identify operon pairs in new, unseen data.** Recall (blue) and specificity (red) for new data sets from (A) *E. coli*, (B) *B. subtilis*, (C) *M. tuberculosis*, and (D) *P. syringiae*. Mean numbers for 100 bootstrapped iterations are shown with 95% confidence intervals (central line in circle).

(i.e. RF and XGB) had the best performance, with XGBoost having almost perfect recall and specificity in this validation set. We then conducted an independent test of our program to understand the broad applicability of our algorithms. We downloaded new RNA-seq data sets from *E.coli* and *B. subtilis*, organisms that were represented in the training data (but this new data is unseen by the algorithm), as well as RNA-seq data sets from *Mycobacterium tuberculosis* (*M. tuberculosis*) and *Pseudomonas syringiae* (*P. syringiae*), organisms (and data) absent from the training data [45–48]. We compared operon calls from our algorithms using these new, unseen data sets against operon annotations from MicrobesOnline. To get a confidence interval for our calls, we sub-sampled 10% of the data with replacement over 100 iterations for each algorithm. These results are plotted along with 95% confidence intervals in Fig 3. There was a range of performance depending on the algorithm used. The GNB and MLP algorithms, for the most part, had higher specificity compared with recall, which suggests that these methods are preferable for conservative operon calls. In many applications, however, we want to capture the largest number of operons. The logistic regression, SVM and tree-based methods (RF and XGB) have higher recall compared with specificity, which allows for a more complete annotation of operons but raises the concern of potential false-positive results. All results were confirmed by plotting receiver operating characteristics (ROC) curves (S2 Fig). The higher recall

and slightly lower specificity bring up the question of whether there may be some operons called by OperonSEQer that are not annotated in MicrobesOnline, which is used as the standard. The question is whether these truly are false-positives or whether we are discovering new operon pairs that have not yet been annotated. Another possible explanation is that a bias in recall and specificity was introduced by variability in the depth and coverage of the sequencing data. Therefore, we analyzed the *M. tuberculosis* data since the various experiments had a large range of sequencing depth (S3 Fig). We found no correlation of total reads, total mapped reads, and percent mapped reads, with recall or specificity, suggesting that depth of sequencing is not limiting when using OperonSEQer.

We compared the OperonSEQer results for *E. coli* and *B. subtilis* with two state-of-the-art methods for operon detection, DOOR and Rockhopper, to ensure that the flexibility of our method did not affect the performance relative to other methods [18,23]. For OperonSEQer, we calculated the recall and specificity for operon calls that were confirmed by 1 to 6 of the algorithms in our method. In other words, we set cutoffs ranging from 1 to 6 for how many algorithms had to call an operon pair before it was considered a true result (S4 Fig). We found that overall, OperonSEQer performs on-par or better than the state-of-the art methods. The heat map in S1 Table shows that with just one of the six algorithms required for calling an operon pair, OperonSEQer has perfect recall for both organisms. There is an expected trade-off between recall and specificity, however, with the compromise point somewhere between 2 and 4 algorithms, depending on the organism. This suggests that using 3 algorithms to call an operon pair is likely a good starting point.

## OperonSEQer enables prediction of new operons

Prior calculations of specificity assume that the operon structure provided by the standard, MicrobesOnline, is ground truth [13]. However, it is possible that the application of RNA-seq data enables prediction of new operons, previously missed by the standard. To address this issue of lower specificity versus novel operons, we sought to corroborate operon calls from OperonSEQer using long-read PacBio SMRTseq transcriptomic data from *E. coli* [25]. In this prior study, a new set of previously unreported operons were discovered based on direct evidence of individual molecules of RNA spanning two genes. We started by comparing each of our individual OperonSEQer algorithms to the gold-standard prediction by MicrobesOnline for confirming SMRTseq calls. We find that on average, the recall and specificity of OperonSEQer matches that of the gold standard, with some models having higher recall and others having higher specificity when surveyed alone. We hypothesized however that the individual models have unique biases, and therefore combining the calls of the models in a manner similar to the results shown in Table 1 (i.e. using OperonSEQer as a suite of algorithms) would allow us to improve on the gold standard result. We compared the performance of our suite of algorithms determining the AUC (Area Under the Curve) for a given number of algorithms in the suite calling an operon pair (eg. 'One' means 1 out of the 6 algorithms called the operon pair, 'Two' means 2 out of 6, and so on). We see that our method performs at an AUC > 0.96 regardless of thresholding when compared with the MicrobesOnline calls as the true positives. On the other hand, if we take the SMRT-seq as the true positives, while the AUC value drops, we find that our suite performs at least as well, if not better in some circumstances than MicrobesOnline, suggesting that with the appropriate method, RNA-seq data alone can be used to obtain better operon prediction than alternative methods that incorporate function and conservation (Fig 4A). We also show this improvement in performance using a Venn Diagram that shows higher overlap of OperonSEQer with SMRT-seq operons, compared with MicrobesOnline (Fig 4B).

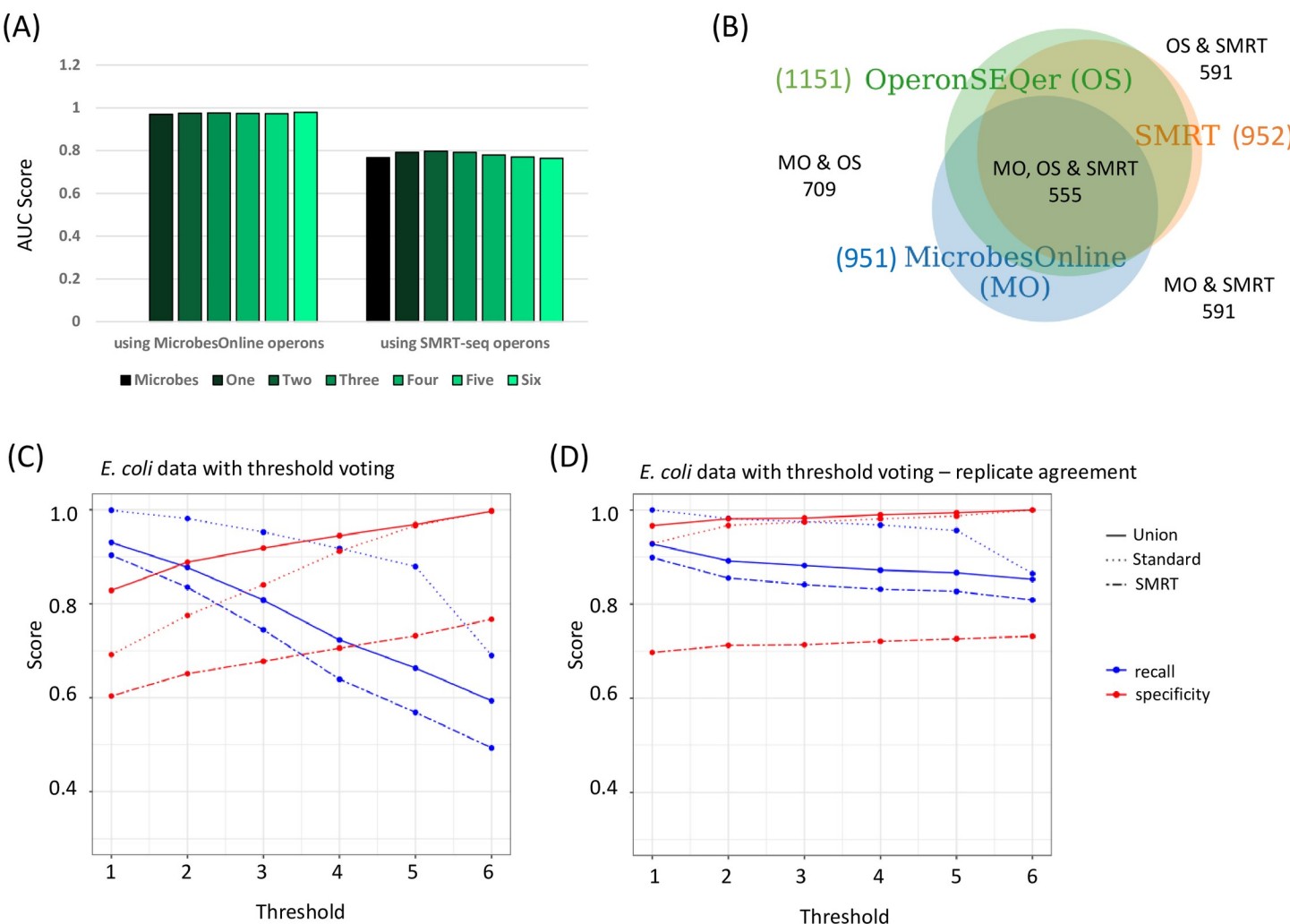

**Fig 4. Operon-SEQer is best used as an ensemble of methods and finds operons not annotated by the standard but detected by PACBIO SMRTseq.** (A) AUC scores of OperonSEQer operon pair calls on MicrobesOnline *E.coli* operons, and both OperonSEQer and MicrobesOnline operon pair calls on SMRT-seq *E.coli* operons. (B) Venn diagrams showing total number of operon calls on *E.coli* genes that pass threshold per method. (C and D) Recall (blue) and specificity (red) of the Operon-SEQer ensemble with algorithm agreement cutoffs of 1–6 for operon pair calls made by the standard (dotted lines), SMRTseq (dashed line), or by the union of calls made by both (solid line); (C) represents all available operon pair data for the new *E. coli* data sets and (D) represents operon pairs that have agreement between two or more replicates.

Next, we sought to assess OperonSEQer's performance on operons called by the standard, by SMRTseq, or by either one. As expected, we see a trade-off between the specificity and the recall of all operon pairs as we increase the number of algorithms required to call an operon pair in *E. coli* (Fig 4C), and this tradeoff exists with data sets for other organisms as well (S4 Fig). Since the SMRTseq data represents only one experimental condition, we do not expect that all operon pairs will be detected with this data set, which is why our method shows lower specificity with SMRTseq-called pairs than with standard-called pairs (Fig 4C). Again, the lower recall with SMRTseq data suggests that some operon pairs with very low expression are detected with long-read sequencing but are difficult to detect with short-read sequencing. The specificity of OperonSEQer is higher (especially at lower algorithm number cutoffs) when we consider all operon pairs called by either SMRTseq or the standard (Fig 4C). This suggests that OperonSEQer is likely detecting operon pairs that are missed by traditional operon callers as previously discussed. A similar result was demonstrated by the authors of Rockhopper, where

they show that some of the operons Rockhopper detects that are not called by the standard can be confirmed by RT-qPCR [18]. Here, we show this on a global scale using long-read sequencing data, and we only require a single experimental condition to achieve this (as opposed to a comparison of multiple experimental conditions).

While OperonSEQer allows for calls from a single experiment, and all our data until now is representative of operon pair calls based on a single RNA-seq result for each gene pair, we tested whether we could use the incidence of RNA-seq replicates (either biological replicates of a single condition or multiple experimental conditions) to strengthen our predictions. We therefore focused only on gene pairs that had data in at least 2 instances of data (i.e. crossed expression thresholds at least twice) and required agreement between the two replicates to make a final call. Replicate agreement was defined as the operon call made for each replicate being the same within an algorithm. We see that requiring two or more calls in agreement drastically improves the recall and specificity for all our comparisons (Fig 4D). Specifically, when we look at operon pairs that are called by either the standard or SMRTseq (solid line in Fig 4D), having even a single algorithm in our set of algorithms call the operon pair ensures a specificity of 96% and a recall of almost 90%, demonstrating that replicates significantly improved the performance of our program without requiring more training.

## Discussion

The emergence of long-read sequencing data has shown us that the discovery of operons in prokaryotes is far from complete. In fact, there are many nuances to operon structure, including modular transcription terminators, that lead to combinations of operons that are difficult to predict based solely on sequence and conservation [25]. While long-read RNA-sequencing is an effective way to address this gap, the limitation with this approach is the need for a wide range of experimental conditions to ensure capture of all operon pairs, which can be time-consuming and costly. As an alternative, we have demonstrated here that the abundance of short-read RNA-sequencing data that has been accumulated over these past decades can be used to discover operon pairs. We show that by using a set of algorithms, we can call operon pairs using short-read sequencing data from a range of organisms with high recall and specificity. In addition, we demonstrate that it is likely that we are identifying non-annotated operon pairs using this method, based on confirmation by long-read sequencing data [25].

Our approach uses a set of algorithms and a threshold voting system, as we found the results both more robust and more flexible compared to individual algorithms. While there are advantages and disadvantages to each approach, the threshold voting system can provide some level of confidence in the call and allows the user to decide whether recall or specificity is more important for their particular needs. In addition, we see that high performance of a single algorithm on one data set does not necessarily guarantee a similarly high performance of that specific algorithm across all data sets, further highlighting the need for a multi-algorithm system to guard against unexpected overfitting of individual algorithms. An example of an ensemble operon caller is CONDOP, which also uses RNA-seq for determining operon gene pairs [49]. The main distinction with our method is that CONDOP requires annotated operons from the DOOR database and outputs a list of condition-specific operons using RNA-seq data based on this previous annotation, while OperonSEQer does *de novo* operon detection using only RNA-seq data and intergenic distance as inputs [49]. We also improve on the methods used by rSeqTU (which uses a combination of random forest (RF) and support vector machine (SVM) models to predict transcriptional units) by incorporating a statistical front-end to allow for more variability across organisms and data sets, and we also use a wide range of training data, as well as multiple ML models and a voting system [17]. We also provide the code required to

re-train our models as data acquisition evolves and novel sequencing data types emerge, which given the statistical front-end transformation, should be broadly applicable. In addition, we have included a feature in the software that allows for stringing together of consecutive operon pairs into multi-gene operons. Other applications in genomics where ensemble methods have proven very useful include annotation of genomic islands, detection of genomic mutations, and gene expression-based phenotype prediction [50–53]. The development of these flexible methods is critical for weathering the natural and technical variation between organisms and data sets, which we can see even between the data sets that we chose to analyze in this study. In addition to flexibility, generalizability has long been an issue with operon calling, with training data often dictating the subset of organisms that can be tested using an algorithm. Our approach circumvents this by taking a gene-agnostic, function-agnostic approach, while simultaneously transforming the data into a statistic and p-value. This allowed OperonSEQer to make calls on organisms and data sets that were unseen during testing with high recall and specificity. In addition, the algorithm can be trained with additional data sets as RNA-seq technology evolves, highlighting the longevity of such an approach.

OperonSEQer has the potential to identify unannotated operon gene pairs that are confirmed by long-read RNA-seq data. This suggests that there are still a number of design rules for operon structure in bacteria that remain unknown, and OperonSEQer can be used as a tool to discover these rules by marking novel operon pairs that are detected through RNA-sequencing but had not previously been identified. We can also ask which of these rules are organism-specific and which are general based on the results of our prediction. There has been a significant amount of work demonstrating that there are a number of dynamic and ever-evolving forces at play when it comes to operon structure, including RNA decay, overlapping transcription and previously uncharacterized functional relationships [2,3,5,54]. Using OperonSEQer, we can survey the large amounts of RNA-seq data that are currently available through public repositories, and we can identify novel operons that can point to new or understudied functions of genes in any prokaryotic organism. Furthermore, since OperonSEQer only requires a single experiment for operon calling, we can compare operon calls between conditions to see whether there are any changes in operon structure based on the state of the cells.

A future goal for OperonSEQer is to incorporate long-read RNA-sequencing as the data becomes available. In fact, OperonSEQer can be consolidated into a larger, modular algorithm that incorporates data from many information streams. It may also be interesting to adapt OperonSEQer for transfer learning for this purpose, as it has been demonstrated that transfer learning can be useful in the generalizability of operon calling [24]. Importantly, our approach of using a statistical method to determine the similarity in expression of different regions of the genome in RNA-seq data, and then using the outputs of this method for machine learning can be applied broadly not only to prokaryotes, but also in understanding regulation of gene expression in higher organisms. Such an endeavor would complement the plethora of work that is currently ongoing in the field of machine learning for understanding gene regulation [55–60]. Ultimately, the key to fully unlocking the potential of machine learning in understanding gene regulation is likely to arise from a combination of computational approaches, with carefully curated and processed data, and methods such as OperonSEQer can be used, adapted, and expanded upon to achieve this goal.

## Materials and methods

### Data sets

For training OperonSEQer, publicly available RNA-seq data were downloaded from Sequence Read Archive (SRA) for *Escherichia coli* (PRJNA274573, PRJNA436580 and PRJNA473128),

*Bacillus subtilis* (PRJNA511580 and PRJNA555096), *Clostridium difficile* (PRJNA244679, PRJNA283975, PRJNA338449 and PRJNA217778), *Burkholderia pseudomallei* (PRJNA413621 and PRJNA312225), *Staphylococcus aureus* (PRJNA514046, PRJNA541911 and PRJNA 546264), *Synechococcus elongatus PCC 7942* (PRJNA315938), *Synechocystis sp. PCC 6803* (PRJNA361291) and *Synechococcus sp. PCC 7002* (PRJNA310120, PRJNA361291 and PRJNA212552).

For testing OperonSEQer, publicly available RNA-seq data were downloaded from SRA for *Escherichia coli* (PRJNA274573, PRJNA436580 and PRJNA473128), *Bacillus subtilis* (PRJNA511580 and PRJNA555096), *Clostridium difficile* (PRJNA244679, PRJNA283975, PRJNA338449 and PRJNA217778), *Burkholderia pseudomallei* (PRJNA413621 and PRJNA312225).

## Preparing, aligning, quantifying and annotating RNA-seq data

RNA-seq data was filtered and trimmed using Trimmomatic for Q-scores > 30, and aligned with Hisat2, and bedtools genomecov was used to extract coverage across the genome [61–63]. A gff3 file corresponding to each organism being surveyed (and matching the genome used for alignment–see S2 Table) was downloaded from Ensembl Bacteria (https://bacteria.ensembl.org/) and filtered for genes only [62]. Coverage was calculated using the bedtools genomcov method with option -d for per-base coverage. In the case of paired-end samples, the -pc option was used to ensure that we obtained the coverage of the interval between the left and right points. Importantly, we next filtered the data for where the mean coverage across at least one gene from the pair of genes being compared is 10 reads, thereby eliminating gene pairs that are not expressed or where no conclusion can be reached. This is an important step in training the algorithm so that it recognizes true negatives and positives and is not side-tracked by regions that are not expressed and therefore cannot be used as predictors.

Following this, we collected pairwise coverage data for adjacent genes, as well as the intergenic region between these genes. With the 5' most gene referred to as gene A and the 3' most gene referred to as gene B, we extract coverage from the 3' 50 bp of gene A (or the whole gene if it is shorter than 50 bp), the central 50 bp of the intergenic region (or the whole intergenic region if it is shorter than 50bp), and the 5' 50bp of gene B (or the whole gene if it is shorter than 50 bp). We performed a Kruskal-Wallis test on pairwise comparisons of coverage or a three-way comparison and recorded the statistic and p-value associated with each test. These, along with the intergenic distance were used as input features for machine learning. Operon calls referred to as 'the standard' were downloaded from MicrobesOnline (www.microbesonline.org/). Long-read SMRT-seq Pacbio data was obtained from doi.org/10.1038/s41467-018-05997-6[25].

## OperonSEQer

OperonSEQer is a set of models with a threshold voting system, and our code is publicly available at https://github.com/sandialabs/OperonSEQer. Briefly, we use the scikit-learn module of Python3 to implement the machine learning algorithms. Algorithms that were used include Logistic Regression with L2 ridge regularization (LR), Support Vector Machine with an RBF kernel (SVM), Random Forest (RF), XGBoost (XGB), Multi-Layer Perceptron (MLP) and Gaussian Naïve Bayes (GNB). Features were scaled for all algorithms except RF and XGB.

The downloaded data was processed as outlined above, and the following features were used for machine learning: length of gene A, length of gene B, intergenic length, Kruskal-Wallis statistics and p-values for pairwise and three-way comparison of gene A, gene B and intergenic coverage (as outlined above), and strand match between gene A and B. The data were

**Table 2. List of hyperparameters for each algorithm used in OperonSEQer.**

| Algorithm | Categorical features | Continuous features |
|---|---|---|
| Logistic regression | Lasso vs ridge regularization | C |
| Random Forest | - | Minimum sample split, maximum depth, number of estimators (all integer) |
| Support Vector Machine | Kernel | C (as applicable), gamma (as applicable) |
| XGBoost | - | Gamma, learning rate, number of estimators (integer) |
| Gaussian Naïve Bayes | - | Variance smoothing |
| Multilayer Perceptron | - | Alpha, Maximum iterations (integer), number of hidden layers (integer), number of neurons per layer (integer) |

scaled (for all relevant algorithms) using MinMaxScalar. Each algorithm's hyperparameters were optimized using Bayesian Optimization (using Gaussian Processes) from GPyOpt methods. The hyperparameters for each algorithm are listed in Table 2.

For the MLP, we used adam as the solver and relu as the activation function. We used only 10 iterations of optimization for all the methods (which we judged as sufficient given high accuracy during optimization), but we provide the code, which can be modified and used to re-optimize hyperparameters in parallel. For each iteration of the optimizer, the model with the current set of hyperparameters was cross-validated 10-fold and the average accuracy of these 10 iterations was used as the metric to evaluate performance. Final validation recall and specificity shown in Table 1.

The model was then saved with the optimized hyperparameters, and new, unseen data from four organisms (two from which we had used alternative data for training, and two from which we had used no data) were used for testing the algorithms. Individual precision and recall values were recorded across each run, with the comparison being made to the 'standard' operons called by MicrobesOnline [13]. In order to obtain confidence intervals for our metrics, we ran a 100-fold bootstrap of subsets of the data sampling 10% at a time. Results were reported as an average of these 100 bootstraps, with 95% confidence intervals calculated from this data. ROC curves and AUC (area under the curve) were calculated using scikit-learn. Calls for n (1–6) number of algorithms were made by tallying the number of times a gene pair got called.

Additional details for OperonSEQer are available at https://github.com/sandialabs/OperonSEQer.

### ROC (receiving operating characteristic) curve analysis

The prediction probability for each OperonSEQer algorithm was calculated in python using with predict_proba function in scikit-learn. False positive and true positive rates were determined using the roc_curve function across a range of probabilities from 0 to 1. AUC (area under the curve) score was determined using the roc_auc_score, with areas closer to 1 being closer to the ideal.

### Supporting information

**S1 Fig. Determining cutoff for average coverage.** Sensitivity (A) and recall (B) tradeoff curves are shown, with the X-axis representing the mean coverage cutoff, the left Y-axis representing the number of data points retained as a result of the cutoff, and the right Y-axis representing

the score. We determined that 10bp was a good cutoff based on the tradeoff between recall, specificity and number of data points.
(TIF)

**S2 Fig. ROC curves for Operon-SEQer performance.** ROC (receiver operating characteristics) curves, and AUC (area under the curve) for the 7 algorithms in Operon-SEQer for the (A) *E. coli*, (B) *B. subtilis*, (C) *M. tuberculosis*, and (D) *P. syringiae* data sets.
(TIF)

**S3 Fig. Number of reads in a data set does not correlate with outcome of Operon-SEQer.** Relationship between recall (blue) and specificity (red) of the 6 algorithms of Operon-SEQer for (A) total reads, (B) total mapped reads, and (C) percent mapped reads in each data set from *M. tuberculosis* (PRJNA521480).
(TIF)

**S4 Fig. Operon-SEQer ensemble tested against new data sets.** Recall (blue) and specificity (red) of the Operon-SEQer ensemble with algorithm agreement cutoffs of 1–6 for operon pair calls for the new data set from (A) *B. subtilis*, (B) *P. syringiae*, and (C) *M. tuberculosis*.
(TIF)

**S1 Table. Comparison of OperonSEQer with DOOR and Rockhopper.** Comparing the recall and specificity of DOOR and Rockhopper with the OperonSEQer ensemble (with agreement of anywhere between 1 and 6 of the algorithms that make up OperonSEQer being used to make operon pair calls). Heat map colors range from yellow (lowest) to white (mid-point) to blue (highest).
(TIF)

**S2 Table. Genomes used for alignment.**
(TIF)

## Acknowledgments

We would like to thank Joshua Podlevsky and Chuck Smallwood for discussions and advice regarding this work, Drew Levin, Bernard Nguyen and Steven Verzi for critical review of the manuscript, and Cameron Kunstadt for testing and troubleshooting of the software package.

## Author Contributions

**Conceptualization:** Raga Krishnakumar, Anne M. Ruffing.

**Data curation:** Raga Krishnakumar.

**Formal analysis:** Raga Krishnakumar.

**Funding acquisition:** Anne M. Ruffing.

**Methodology:** Raga Krishnakumar.

**Project administration:** Anne M. Ruffing.

**Software:** Raga Krishnakumar.

**Validation:** Raga Krishnakumar.

**Writing – original draft:** Raga Krishnakumar.

**Writing – review & editing:** Raga Krishnakumar, Anne M. Ruffing.

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
