## [Decision Letter · Decision Letter 0]

20 Sep 2021

Dear %TITLE% Krishnakumar,

Thank you very much for submitting your manuscript "OperonSEQer: A set of machine-learning algorithms with threshold voting for detection of operon pairs using short-read RNA-sequencing data" for consideration at PLOS Computational Biology.

As with all papers reviewed by the journal, your manuscript was reviewed by members of the editorial board and by several independent reviewers. In light of the reviews (below this email), we would like to invite the resubmission of a significantly-revised version that takes into account the reviewers' comments.

We cannot make any decision about publication until we have seen the revised manuscript and your response to the reviewers' comments. Your revised manuscript is also likely to be sent to reviewers for further evaluation.

Sincerely,

Nicola Segata

Associate Editor

PLOS Computational Biology

Ilya Ioshikhes

Deputy Editor

PLOS Computational Biology

Reviewer's Responses to Questions

**Comments to the Authors:**

Reviewer #1: Review in attachment as pdf

In this paper, the authors presented OperonSEQer, a new tool for de-novo operon prediction in prokaryotic genomes using RNA-seq data. Coverage values of gene pairs and their intergenic regions are compared through non-parametric analysis of variance. Then the test metrics are used as input in a wide set of machine learning (ML) models thus ensuring a flexible approach, not considering species, genes nor function. This approach was trained, tested and validated on a significant set of diverse bacterial species for which RNA-seq data was available. Interestingly, the use of a set of predictions from different ML models allowed the authors to implement a “threshold voting system” based on the agreement between models and offering the opportunity for users to choose a setup favoring high recall or high precision based on agreement between models.

Moreover, the authors had a relevant critical view on the data they used, especially the database of operons used as truth in the training of the models and the evaluation part. By comparing this database with their results as well as with SMRTseq transcriptomics data (long read transcriptomics) they were able to confirm that OperonSEQer can identify previously unidentified operons.

Overall, I think this paper can be of interesting value for the bacterial genomics field. First, the application of the software could be of high impact for novel operon detection and a better understanding of bacterial genomes organisation and function. The multi-model approach provides an interesting and flexible way of analyzing and discussing the results. Finally, the relevant use of publicly available data on top of the code available on a public GitHub repository ensures the reproducibility and transparency of the work.

However, the manuscript has some clarity and precision issues, especially regarding the non-parametric analysis of variance part and the assessment of novel operon detection using SMRTseq data. Indeed, some parts of the main text can remain unclear and could be easily improved by adding more details.

---Major issues---

Analysis of variance (Kruskal-Wallis test) : The features used in the ML models are described as the statistic and p-value of 3-way Kruskal-Wallis test (coverages of gene A, B and their intergenic region ) as well as each 2-way test. In that case post-hoc pairwise tests might be more appropriate than 2-way KW tests as seen in the code (maybe Dunn’s test instead). Moreover, multiple testing correction is not mentioned in the manuscript and should be needed for such analysis.

Mapping RNA-seq read on genome : As explained in the Method part, RNA-seq data are aligned across genomes and filter for genes only with a gff3 file retrieved from Ensembl Bacteria. As one could expect intra-species differences in terms of genome composition and operon presence/absence, the mention of the genomes used as reference for this analysis are absolutely needed since results will be impacted by the choice of a (set of) reference genome.

Prediction of new operon : Clarity issue in the paragraph ranging from line 235 to 259 in the manuscript as well as the Fig 4.A that is described in here. It is difficult to understand what the “fraction” represents. Especially in the way data should be interpreted. It is “>80% of the operon calls made by Operon-SEQer are confirmed by standard and SMRT-seq data '' while the manuscript (l. 247-248) seem to claim that when standard and SMRT seq agree, >80% of these calls can be found by Operon-SEQer. Same remarks for the end of the paragraph (l. 250-254).

In a nutshell, the clarification between “X% of operons confirmed by Y are found by OperonSEQer” and “X% or the operons found by OperonSEQer are confirmed by Y” would improve the explanation of this analysis and help support the proof of new operons discovery.

It would also be interesting and help the understanding to have absolute values available here on top of relative ones.

---Minor issues---

in Author Summary

* l. 41-42 : distinction between the operon (the organization of the genetic information) and the polycistronic RNA (the molecule conveying it)

in Results

* Results l. 156 : What’s correlating is the likelihood (or another metric) of a pair of genes being in an operon and not the pair of genes.

* l. 193 Figure 2b is mentioned but lacking in the provided manuscript

* l. 214 Very strange transition between sentences that seem to talk about different issues

* l. 226-227 Heatmap is mentioned as Supp. Fig. 3 while being Supp. Table 1 in the provided manuscript

* The difference between Figures 4C and 4D could be highlighted more, on the figure, the Title changes from “E. coli data” to “E. coli data with thresholds voting” while both figures display the impact of threshold voting.

In Discussion

* l. 305 Missing reference ?

in Material and Methods

* l. 415 Table in the main text without title or number

Misc

* References 21 to 37 not included in the main text

Reviewer #2: The manuscript by Raga Krishnakumar and Anne M. Ruffing describes a ensamble of machine learning algorithms "OperonSEQer" that use RNA-seq data to call operon-pairs.

My review will focus on the practical perspective as I'm not an expert in the Machine Learning models applied in the manuscript. I found the manuscript pleasing to read and liked the step-by-step reporting on features that made it easy to follow the authors reasoning.

Specific comments:

- It could be made more clear if the authors are looking at operon-pairs or operons in general (which can constitute more than 2 genes). Does the software automatically extract potential larger operons? I assume that the information would be readily available by looking up the operon pairs, but for non-expert users it might be a convinient feature to have.

- MicrobesOnline is used as the "gold standard". However, as a non-expert I would like a brief description of how MicrobesOnline have called those operons in the first place.

- line24: "high coverage" of operons could be confused in this context. I guess what you mean is high sensitivity or high specificity?

- Line385: It's mentioned that 10 reads for at least one gene is used to avoid false calls? Why was 10 reads chosen? From the text it is also a little unclear to me if it is 10 reads, or an average coverage of 10 that it is used. It would be relatively easy to subset the data to simulate the impact of coverage on the calling of operon-pairs.

- line 389: Why was coverage only collected in a single 50 bp window at 3', middle and 5'? Why not use the entire gene? Anecdotal I remember experiements where there was quite some bias in coverage between the 3' and 5' of genes. If it is really only 50 bp from each element that is used, then it should also be added to Figure 1 to explain how the method works.

Figure 1: While I do like conceptual figures, I think it would be much more impactful to take 3 real examples from e.g. E. coli with real gene-names, lenghts and coverages.

Figure 3: What does the confidence intervals show? As you argue the largest difference would be different conditions and experimental batch effects. Hence, I'm not sure what the confidence interval here is ment to show?

Figure 4: I guess it was only Ecoli where SMRT data was available for? Could be written explicit in the figure text.

- "SMRT only" has low rate of confirmed calls -> was this due to low coverage of SMRT seq or very specific conditions?

- A little confusing with the naming of SMRT "no" and "yes". As I understand it SMRT-No is the MicrobesOnline call (called "stand" in figure 4?), while SMRT-Yes is MicrobesOnline+SMRT.

Reviewer #3: In this manuscript, Raga Krishnakumar and Anne M. Ruffing describe a new approach to predict, from a single RNA-Seq data set, pairs of adjacent genes that are co-transcribed (i.e. within the same operon). The proposed approach, OperonSEQer, uses different “machine learning” approaches (classifiers) to combine information on the comparison between expression signal of the last 50 bp of the first gene (upstream) in the pair, of the intergenic region (up to 50 bp), of the first 50 bp of the second gene (downstream), and also on the length of the intergenic region. The rationale is that genes in a same operon are transcribed at the same level and are separated by a short intergenic region. Comparison between expression levels relies on statistics derived from the Kruskal-Wallis test. The classifiers are trained using operon information from the MicrobesOnline database. Performance are compared to two previously proposed methods (Rockhopper and DOOR) and a simple voting strategy is proposed to combine the results of the different classifiers.

As tackled in this work, the problem of operon prediction is fairly simple excepted for reasons stemming from the notion of operon which does not reflect the full complexity of bacterial transcriptomes (transcription units can overlap and transcription termination be partial). In this context, the proposed methods appears as ad-hoc solutions that are simple and work (even if maybe not well theoretically grounded, see major comments below). It is not clear that they outperform previously proposed approaches. For these reasons, my general feeling is that the manuscript is of limited originality and interest.

Major comments:

* The introduction states the problem of operon prediction without mentioning the fact that the notion of pairs of genes in operon or not cannot describe the full complexity of bacterial transcriptomes. My personal opinion is that the problem tackled in this work is only of limited interest.

* The introduction does not provide enough insights on the principles underlying the different operon predictions methods already described in the literature over the years (from co-transcription, to intergenic distance and prediction of intrinsic terminators...).

* The text seems often superficial or trivial.

* L201-202 I do not understand the use of the subsampling approach to establish confidence intervals (a bootstrap would be more appropriate). I did not find more details in the M&M section.

* L204-211 I do not understand the interest of these comments about the difference in recall vs. accuracy trade-offs between the different classifiers when this trade-off can simply be tuned by changing the cut-off on the probability of the prediction (as done to draw the ROC curves in Supp Figure 1). The AUC is probably the appropriate measure to compare the classifiers in this context.

* L220-231 The authors show that by combining the results of the different classifiers with a simple voting scheme they can modify the recall vs. accuracy trade-off but they do not demonstrate that this allow to outperform (in terms of AUC) the best of their classifiers. In this context, the statement made L307-308 does not seem rigorously supported by the results presented.

* As presented I do not fully grasp the point analysis of SMRT-seq operon pairs. The authors seem to put forward the idea that their method is interesting because it can make true predictions (i.e. validated by some data sets) that are not in the standard lists of operons. This is not really surprising and is probably expected for any reasonable de novo method given the complexity of the bacterial transcriptome.

* The material and methods is not described with enough details. For instance it is not clear how paired-end reads are handled in the bioinformatic workflow and if the distribution of counts that are compared between regions are the numbers of reads starting at each position or the numbers of reads overlapping each position within each of the three regions (with which option genomecov is used?). Importantly in the second case (overlapping reads or fragments) the use of Kruskal-Wallis test does not seem theoretically justified since the length of the reads or fragments induces an important correlation between counts.

* Given the typical short length of the intergenic region and the typical fragment length (nowadays around 300 bp for pair-end “short-read” datasets?) I do not fully understand why the authors does not simply examine whether there are reads that overlap the two adjacent genes in order to see if they are co-transcribed.

Some minor comments:

* Spelling of the name of the proposed program is not always consistent (OperonSEQer or Operon-SEQer)

* The text contains typos (“an technological opportinity” L79, “resoureces” L138, ...)

* L92 it is a mistake to oppose “non-parametric” and “normal”. Many parametric statistical methods have been developed to analyze data from distributions that are not normal.

* L225 does the authors mean Supp Table 1 instead of Supp Figure 3?

* L276-277 analysis of single condition should not be presented as the opposed of a comparison between two experimental conditions. Many data sets contains much more than two experimental conditions...

**Have the authors made all data and (if applicable) computational code underlying the findings in their manuscript fully available?**

Reviewer #1: Yes

Reviewer #2: Yes

Reviewer #3: None

PLOS authors have the option to publish the peer review history of their article (what does this mean?). If published, this will include your full peer review and any attached files.

Reviewer #1: No

Reviewer #2: **Yes: **Mads Albertsen

Reviewer #3: No
---

## [Decision Letter · Decision Letter 1]

7 Dec 2021

Dear %TITLE% Krishnakumar,

We are pleased to inform you that your manuscript 'OperonSEQer: A set of machine-learning algorithms with threshold voting for detection of operon pairs using short-read RNA-sequencing data' has been provisionally accepted for publication in PLOS Computational Biology.

Best regards,

Nicola Segata

Associate Editor

PLOS Computational Biology

Ilya Ioshikhes

Deputy Editor

PLOS Computational Biology

Reviewer's Responses to Questions

**Comments to the Authors:**

Reviewer #1: The authors revised the manuscript and addressed the previous comments. The main results are now clearer and the improvement in the operon prediction proposed by the tool is presented in a better way.

Minor comment

Fig 4B could be transformed into UpSet plot that could highlight the concording results between the different methods

Reviewer #2: My comments have been adressed nicely, and I do not have any additional comments.

**Have the authors made all data and (if applicable) computational code underlying the findings in their manuscript fully available?**

Reviewer #1: Yes

Reviewer #2: Yes

PLOS authors have the option to publish the peer review history of their article (what does this mean?). If published, this will include your full peer review and any attached files.

Reviewer #1: No

Reviewer #2: No

---

## [Editor Report · Acceptance letter]

3 Jan 2022

PCOMPBIOL-D-21-01407R1 

OperonSEQer: A set of machine-learning algorithms with threshold voting for detection of operon pairs using short-read RNA-sequencing data

Dear Dr Krishnakumar,

I am pleased to inform you that your manuscript has been formally accepted for publication in PLOS Computational Biology. Your manuscript is now with our production department and you will be notified of the publication date in due course.

With kind regards,

Olena Szabo
